# Influence of Adherence to the Mediterranean Diet on Vascular Health and Healthy Aging in Women: Assessment Using CAVI

**DOI:** 10.3390/nu17182963

**Published:** 2025-09-16

**Authors:** Inés Llamas-Ramos, Rocío Llamas-Ramos, María Cortés Rodríguez, Emiliano Rodríguez-Sánchez, Manuel A. Gómez-Marcos, Marta Gómez-Sánchez, Leticia Gómez-Sánchez

**Affiliations:** 1Faculty of Nursing and Physiotherapy, Universidad de Salamanca, 37007 Salamanca, Spain; inesllamas@usal.es (I.L.-R.); rociollamas@usal.es (R.L.-R.); 2Instituto de Investigación Biomédica de Salamanca (IBSAL), 37007 Salamanca, Spain; emiliano@usal.es; 3Primary Care Research Unit of Salamanca (APISAL), Health Centre of San Juan, Av. Portugal 83, 2º P, 37005 Salamanca, Spain; martagmzsnchz@gmail.com (M.G.-S.); leticiagmzsnchz@gmail.com (L.G.-S.); 4University Hospital of Salamanca, Health Service of Castile and Leon (SACyL), 37007 Salamanca, Spain; 5Department of Statistics, University of Salamanca, 37008 Salamanca, Spain; 6Department of Hematology, University Hospital of Salamanca, 37008 Salamanca, Spain; 7Primary Healthcare Management, Castilla y León Regional Health Authority (SACyL), 37007 Salamanca, Spain; 8Department of Medicine, University of Salamanca, 37007 Salamanca, Spain; 9Red de Investigación en Cronicidad, Atención Primaria y Promoción de la Salud (RICAPPS), 37005 Salamanca, Spain; 10Home Hospitalization Service, Marqués of Valdecilla University Hospital, s/n, 39008 Santander, Spain; 11Emergency Service, University Hospital of La Paz, P. of Castellana, 261, 28046 Madrid, Spain

**Keywords:** vascular aging, arterial stiffness, CAVI, mediterranean diet, cardiovascular risk, lifestyle

## Abstract

**Background:** Vascular aging, characterized by a progressive increase in arterial stiffness, is associated with an increased risk of cardiovascular disease. The Mediterranean diet (MD) has been shown to have protective effects on cardiovascular health, but its relationship with vascular aging, as assessed by the Cardio-Ankle Vascular Index (CAVI), is not yet well established. **Objective:** To analyze the association between adherence to the MD and vascular aging estimated with CAVI in a sample of Spanish women with no history of previous cardiovascular disease. **Methods:** A cross-sectional study was conducted in 1468 women (aged 30–80 years), classified into three categories according to CAVI: healthy vascular aging (HVA), normal (NVA), and early (EVA). Adherence to the MD was assessed using the MEDAS questionnaire validated in Spain. ANOVA and chi-square tests were used to compare baseline characteristics, and multinomial logistic regression was used to analyze the association between MD adherence and vascular aging status, adjusting for age, sex, and drug treatment. **Results:** The CAVI increased progressively with age (*p* < 0.001) and was associated with higher blood pressure, dyslipidemia, hyperglycemia, and diabetes mellitus. Adherence to the MD was significantly higher in groups with lower arterial stiffness (*p* < 0.001). In the multivariate analysis, greater adherence to MD was associated with a higher probability of healthy vascular aging compared to NVA (OR: 1.872; 95% CI: 1.366–2.566; *p* < 0.001) and EVA (OR: 1.738; 95% CI: 1.363–2.218; *p* < 0.001). **Conclusions:** Greater adherence to the MD is associated with a healthier vascular aging profile. Promoting this dietary pattern may be an effective strategy for preventing arterial stiffness and reducing the cardiovascular risk associated with aging.

## 1. Introduction

Aging brings with it a series of physiological changes linked to the passage of time [1]. Among these, vascular aging and cardiovascular disease have a high prevalence in women, estimated at 275.2 million, and an age-standardized global prevalence of 6403 per 100,000 people per year [2]. In addition, during menopause, women experience a decrease in estrogen levels, which is important in cardiovascular disease and mortality [3].

Vascular aging is characterized by a progressive increase in arterial stiffness, a phenomenon closely related to the development of cardiovascular diseases [4,5]. Arterial stiffness is considered one of the main subclinical markers in the progression of atherosclerosis [6]. The latter, common to most cardiovascular diseases, manifests itself through the accumulation of lipids, inflammatory processes, and endothelial dysfunction, which can promote the formation of fatty streaks in the vascular wall and the subsequent rupture of atheromatous plaque [7]. In this context, the Cardio-Ankle Vascular Index (CAVI) has established itself as a useful index for assessing arterial stiffness along the arteriovascular axis from the aorta to the ankles, considering it a measure of central and peripheral arterial stiffness. It is considered a reliable marker of vascular aging [8]. Several studies have shown that, as age increases, the elasticity of the arteries decreases and stiffness increases, which is reflected in a progressive increase in CAVI in older people. Therefore, this index is sensitive not only for monitoring the progression of cardiovascular disease, but also for evaluating the effectiveness of therapeutic interventions [8].

On the other hand, some recent studies have pointed out that a low-carbohydrate diet, with moderate fat intake and high protein intake, can significantly contribute to weight loss in postmenopausal women, thus promoting better overall health and preventing chronic diseases [9]. In this regard, it is essential to identify appropriate dietary patterns to prevent weight gain in this population. One of the most recommended nutritional strategies is the Mediterranean diet (MD), characterized by its low carbohydrate content and widely supported by scientific evidence [10]. Since the landmark Seven Countries Study began in the 1960s, the MD has gained recognition by analyzing the relationship between dietary patterns and cardiovascular health globally [11]. This diet promotes high consumption of plant-based foods (fruits, vegetables, legumes, whole grains, and nuts), monounsaturated fats from extra virgin olive oil, and moderate consumption of dairy products and fish, as well as reduced intake of red meat [10]. In addition, it has been associated with a decrease in the prevalence of chronic diseases, such as colorectal cancer [12].

The MD has been shown to have cardioprotective effects such as improved lipid profile, vascular function, and blood pressure, as well as reduced oxidative stress and inflammatory biomarkers that can generate antioxidant and antiatherosclerotic effects [13,14]. However, nutrition must be considered from different perspectives depending on gender. Studies should be conducted in female populations [15], as women may have previous risk factors such as preeclampsia or gestational diabetes during pregnancy, polycystic ovary syndrome, inflammatory diseases such as systemic lupus erythematosus, or rheumatoid arthritis [16]. Furthermore, although the evidence is limited, diet may be influenced by hormones and different nutrient metabolisms [15]. Therefore, the aim of this study was to analyze the association between adherence to the MD and vascular aging estimated with CAVI in a sample of Spanish women with no history of previous cardiovascular disease.

## 2. Materials and Methods

### 2.1. Study Design

Cross-sectional study grouping women participating in the EVA [17], MARK [18], and EVIDENT [19] studies. All studies have been approved by the ethics committee and are registered in clinical trials under the following codes: EVA: NCT02623894; EVIDENT: NCT02016014; and MARK: NCT01428934.

### 2.2. Study Population

Women aged between 35 and 75 years old participated in the three studies conducted in primary care centers: EVA [17], MARK [18], and EVIDENT [19]. Random sampling was performed among women attending primary care consultations. Figure 1 shows the participants included and excluded from each study.

### 2.3. Ethical Considerations

The three studies included in this manuscript were approved by the relevant ethics committee under registration numbers PI15/01039 and PI20/10569 for the EVA study; PI10/02043 for the MARK study; and PI83/06/2018 for the EVIDENT study. The standards established in the Declaration of Helsinki and the guidelines established by the WHO for the development of observational studies were followed. All participants were informed and signed an informed consent form when they agreed to participate. In addition, the standards established in Organic Law 3/2018, of 5 December, on the Protection of Personal Data and guaranteeing rights, and Regulation (EU) 2016/679 of the European Parliament and of the Council of 27 April 2016, on Data Protection (GDPR) for data confidentiality were followed.

### 2.4. Variables and Measuring Instruments

#### 2.4.1. Vascular Aging

Vascular aging was classified into healthy aging (HVA), normal aging (NVA), and early aging (EVA) based on the cut-off points established by the EVA Group for women estimated using CAVI [20]. Women with CAVI values at or below the 25th percentile were categorized as HVA, those falling between the 25th and 75th percentile as NVA, and those above the 75th percentile as EVA. In line with previous studies [21,22,23], women with a diagnosis of type 2 diabetes who were initially classified in the HVA group were reclassified as NVA [24]. This decision is supported by the fact that type 2 diabetes is typically diagnosed after several years of asymptomatic evolution, during which vascular damage often occurs. Therefore, it is unlikely that individuals with type 2 diabetes truly present a profile consistent with “healthy vascular aging”. This reclassification aims to avoid mislabeling women with an established metabolic condition associated with vascular alterations as having HVA. Figure 2 shows the percentage of women in each group.

#### 2.4.2. Arterial Stiffness

The measurement of CAVI was performed with the VaSera VS-1500 device (Fukuda Denshi Co., Ltd., Tokyo, Japan) [25]. In the assessment protocol, the patient remained at rest for 10 min before starting the measurement and had not smoked or consumed caffeine 1 h before the test. The cuffs were adjusted to the circumference of their arms and legs. A microphone was positioned over the sternum at the second intercostal space, while electrodes were placed on both arms and legs. The CAVI measurement was considered valid when obtained during at least 3 consecutive heartbeats. To obtain the CAVI values, the following equation was used, substituting the stiffness parameters β = 2 ρ × 1/(Ps − Pd) × ln (Ps/Pd) × PWV2 where ρ is blood density, Ps and Pd are SBP and DBP in mmHg, and PWV was measured between the aortic valve and the ankle [25].

#### 2.4.3. Mediterranean Diet

The MEDAS questionnaire used in the PREDIMED study was selected to assess the MD. This 14-item questionnaire has been validated in the Spanish population [26]. The questionnaire consists of 14 items, of which 12 assess the frequency of consumption of specific food groups and 2 address eating habits characteristic of the Spanish population. Each item is scored dichotomously (0 or 1), yielding a total score between 0 and 14 points. Women with scores above the median were classified as adherent to the MD. A value of 1 point was assigned to the following behaviors: using olive oil as the main culinary fat; consuming at least four tablespoons of olive oil daily (1 tablespoon = 13.5 g); eating two or more portions of vegetables; eating three or more pieces of fruit; limiting intake to less than one serving of red or processed meat, less than one serving of animal fats, and less than one cup (100 mL) of sugar-sweetened beverages; and preferring white over red meat. Additionally, adherence was reinforced by awarding 1 point for consuming seven or more glasses of wine per week, three or more servings of legumes, three or more servings of fish, three or more servings of nuts or dried fruits, at least two servings of sofrito, and fewer than two servings of baked goods [26].

#### 2.4.4. Descriptives

As published in the protocols of the included studies [17,18,19], blood pressure, height, and weight measurements were taken in primary care units.

The tests were performed between 8:00 and 9:00 a.m. after a 12 h fast and were processed in reference laboratories.

Blood pressure values ≥ 140/90 mmHg or taking antihypertensive drugs were considered a diagnosis of hypertension; having fasting plasma glucose values ≥ 126 mg/dL or HbA1c ≥ 6.5% or taking hypoglycemic agents was considered a diagnosis of type 2 diabetes mellitus; having fasting total cholesterol values ≥ 240 mg/dL, low-density lipoprotein cholesterol (LDL-C) ≥ 160 mg/dL, high-density lipoprotein cholesterol (HDL-C) ≤ 40 mg/dL in men and ≤50 mg/dL in women, or triglycerides ≥ 150 mg/dL, or taking lipid-lowering drugs was considered a diagnosis of dyslipidemia. A BMI ≥ 30 kg/m^2^ was used to define obesity [17,18,19].

### 2.5. Statistical Analysis

Continuous variables are presented as mean ± standard deviation (SD), while categorical variables are reported as absolute frequencies and percentages. To compare baseline characteristics between age groups and according to the degree of vascular aging (HVA, NVA and AVA), analysis of variance (ANOVA) was used for continuous variables and the chi-square test (χ^2^) for categorical variables.

Likewise, a multinomial logistic regression analysis was used to analyze the association between MD adherence (independent variable) and vascular aging status (dependent variable with three categories: HVA, NVA, and AVA), using the HVA group as a reference. The model was adjusted for age, sex, and use of antihypertensive, hypoglycemic, and lipid-lowering drugs. SPSS Statistics for Windows, version 28.0 (IBM Corp., Armonk, NY, USA) was used. A *p*-value < 0.05 was considered the threshold for statistical significance.

## 3. Results

### 3.1. Baseline Characteristics

The study sample included a total of 1468 women. Significant differences were observed between age groups in almost all variables analyzed (*p* < 0.05). Regarding MD, the mean total score was 6.03 ± 1.98, with the highest values in the younger age groups, particularly in the 30–40 age group (6.74 ± 0.24), and a progressive decrease up to the 60–70 age group (5.87 ± 0.08), although with an uptick in the 70–80 age group (6.27 ± 0.13). Overall, 59.9% of participants showed adequate adherence to MD, which was less frequent in younger groups (*p* = 0.001).

With regard to conventional cardiovascular risk factors, a progressive increase in systolic blood pressure was found with age, from 102.73 ± 2.31 mmHg in the 30–40 age group to 135.91 ± 1.24 mmHg in the 70–80 age group (*p* < 0.001). Diastolic blood pressure showed an inverse trend, peaking in the 50–60 age group (82.23 ± 0.50 mmHg) and subsequently decreasing. The prevalence of hypertension was high across the entire cohort (89.2%), although it was particularly high in the 30–40 age group (98.6%) and the 40–50 age group (93.7%) (*p* = 0.025). The arterial stiffness index measured by CAVI increased progressively and significantly with age, from 6.58 ± 0.12 m/s in the youngest group to 9.29 ± 0.07 m/s in those over 70 years of age (*p* < 0.001), reflecting the functional deterioration of the arterial tree associated with aging (Table 1).

### 3.2. Characteristics of the Subjects Included with and Without Healthy Vascular Aging

Women were classified into healthy vascular aging (HVA, n = 254), normal aging (NVA, n = 746), and early vascular aging (EVA, n = 468). We found a better score for MD adherence in the HVA group (6.34 ± 0.12) compared to the EVA group (5.72 ± 0.09) (*p* < 0.001). On the other hand, the proportion of subjects with high adherence to the MD was higher in the EVA group (69.0%) than in the HVA group (34.3%), which could reflect greater implementation of dietary changes in subjects with higher cardiovascular risk (*p* < 0.001).

In terms of cardiovascular risk factors, the EVA group had significantly higher systolic (133.54 ± 0.94 mmHg) and diastolic (81.84 ± 0.50 mmHg) blood pressure readings compared to the other groups (*p* < 0.001). Finally, the arterial stiffness index, measured by CAVI, was significantly lower in the HVA group (6.95 ± 0.06 m/s) and increased progressively in the NVA (8.19 ± 0.03 m/s) and EVA (9.44 ± 0.04 m/s) groups (*p* < 0.001), validating its usefulness as an objective marker of vascular aging (Table 2a). In addition, pairwise *p*-values for differences in total cholesterol, triglycerides, fasting plasma glucose (FPG), and HbA1c between subjects with HVA, NVA, and EVA are shown in Table 2b.

### 3.3. Association Between CAVI and MD Overall and by Sex Multinomial Logistic Regression

Figure 3 shows the values of the multinomial logistic regression analysis. The results showed that greater adherence to the MD was significantly associated with a higher probability of healthy vascular aging. Specifically, individuals with greater adherence to the MD were 1.87 times more likely to belong to the HVA group compared to the NVA group (OR: 1.872; 95% CI: 1.366–2.566; *p* < 0.001). Likewise, the probability of belonging to the HVA group versus the EVA group was 1.74 times higher in those who adhered more closely to the MD (OR: 1.738; 95% CI: 1.363–2.218; *p* < 0.001).

## 4. Discussion

In the present study, a significant association was identified between greater adherence to the MD and lower arterial stiffness, as measured by the CAVI, supporting its protective role in the context of vascular aging in the female population. Our results indicate that greater adherence to the MD is associated with a significantly higher probability of presenting with HVA. This finding is consistent with previous research that has documented the beneficial effects of dietary patterns characterized by high consumption of plant-based foods, unsaturated fats, and antioxidant compounds on improving endothelial function and preserving arterial elasticity [14,27,28].

It is important to note that vascular aging differs between the sexes. In women, arterial stiffening and endothelial dysfunction can accelerate not only with advancing age and during the menopausal transition, but also in the context of specific conditions such as polycystic ovary syndrome or adverse pregnancy outcomes, which significantly increase the risk of developing cardiovascular disease [29,30,31]. Similarly, diseases such as lupus erythematosus contribute to increased cardiovascular risk in women, while healthy eating habits may help to control it [32,33]. To understand the structural and functional changes in the arterial wall, including endothelial dysfunction, loss of elasticity, and vascular remodeling, the CAVI has emerged as a useful tool for assessing arterial stiffness independently of blood pressure at the time of measurement [34,35]. This fact is supported by Namekata et al. [25] and Shirai et al. [36], in their studies proposed the CAVI as a novel, non-invasive indicator of arterial stiffness that reflects the functional and structural properties of the arterial wall independently of blood pressure at the time of measurement. Namekata et al. established baseline reference values for CAVI in a large cross-sectional study, supporting its use as a standardized criterion for the evaluation of arteriosclerosis in diverse populations [25]. In addition, Shirai emphasized the clinical utility of CAVI for analyzing vascular function and its role as a reliable tool in cardiovascular risk assessment [36]. These studies provide a solid foundation for the use of CAVI as the main measure of vascular aging in the present research. The results of this article showed a progressive increase in CAVI with age, confirming that arterial stiffness increases with aging. This pattern was accompanied by an increase in systolic blood pressure, dyslipidemia, and impaired glucose metabolism in the older age groups, elements that, taken together, constitute a higher cardiovascular risk profile and are also reflected in more accelerated vascular aging. A relevant finding of our analysis was that, although individuals with EVA had a higher frequency of adherence to MD, their CAVI values were higher. This could be explained by the recent adoption of dietary changes in people already diagnosed with risk factors or disease, which underscores the importance of implementing preventive interventions early on.

The association between MD and arterial stiffness observed in this study is consistent with multiple epidemiological studies and clinical trials that have shown improved endothelial function, reduced inflammatory markers, and a protective effect on vascular integrity with adherence to DM [37,38,39]. For example, the PREDIMED trial demonstrated that adherence to the MD supplemented with extra virgin olive oil or nuts is associated with a significant reduction in cardiovascular risk [40,41]. Similarly, Frank et al. [42] observed in their study of 55 subjects that following this dietary pattern not only optimizes the lipid and metabolic profile but could also favorably influence early markers of subclinical vascular and cardiac damage. On the other hand, consumption of steamed chicken breast as a protein-rich food together with resistance training could counteract increased stiffness in older women [43]. In addition, a recent study conducted by Zupo et al. [44] evaluated 128 publications and selected 16 observational and randomized controlled trials. They concluded that people who follow the MD have less arterial stiffness and, therefore, better cardiovascular health. However, they pointed out that there are multifactorial biological pathways that still need to be corroborated.

One thing to keep in mind is that in this study, the HVA group, although with a lower prevalence of comorbidities such as diabetes or dyslipidemia, had higher BMI and obesity values, which should be taken into account when implementing treatment strategies. This finding was corroborated in studies such as that by Hauner et al. [45], who analyzed the effect of a telephone-based lifestyle intervention program on reducing body weight and waist circumference, decreasing cardiovascular risk factors, and improving lifestyle. They found that a lifestyle intervention program, delivered primarily by telephone, can reduce body weight and waist circumference, improve diet quality, and decrease cardiometabolic risk in overweight or obese women with intermediate to high-risk breast cancer. In addition, weight loss, waist circumference reduction, and improved eating habits were maintained up to two years after the intervention [45].

Another paradoxical finding of our study was that, although women in the HVA group showed higher mean MD adherence scores, a greater proportion of women in the EVA group were classified as having high adherence. This discrepancy may be explained by differences in the distribution of adherence scores across groups. In particular, women with EVA may be more likely to adopt substantial dietary changes after becoming aware of their higher cardiovascular risk, which could increase the proportion surpassing the “high adherence” threshold despite lower overall scores. Additionally, the reliance on self-reported dietary questionnaires could have introduced recall or reporting bias, particularly among participants with greater health concerns, potentially contributing to this finding. Future longitudinal studies with objective dietary assessment methods are needed to clarify whether this pattern reflects true behavioral change or methodological limitations.

Among the strengths of the present study are the use of the CAVI as a standardized and reproducible measure of arterial stiffness, as well as the large sample size and the inclusion of a population with a wide age range. Nevertheless, some important limitations should be acknowledged. First, the cross-sectional design precludes establishing any causal inference; therefore, the associations observed should be interpreted as correlations rather than cause–effect relationships. Second, adherence to the MD was assessed using self-reported dietary questionnaires, which are inherently prone to recall and reporting bias, and may not fully capture actual dietary intake. These limitations should be considered when interpreting our findings, and future longitudinal studies with objective dietary assessment methods are warranted to confirm these results.

## 5. Conclusions

The results of this study show that greater adherence to the MD is associated with a healthier vascular aging profile, characterized by lower central and peripheral arterial stiffness as assessed by the CAVI. These findings reinforce the protective role of this dietary pattern in preventing age-related vascular deterioration, beyond conventional risk factors such as hypertension, dyslipidemia, or diabetes mellitus.

The implementation of nutritional intervention strategies based on the MD could effectively contribute to promoting healthy cardiovascular aging and reducing the risk of chronic diseases in the general population. However, longitudinal studies are needed to confirm the causal direction of this association and determine the long-term impact of sustained adherence to this dietary pattern on vascular health.

## Figures and Tables

**Figure 1 nutrients-17-02963-f001:**
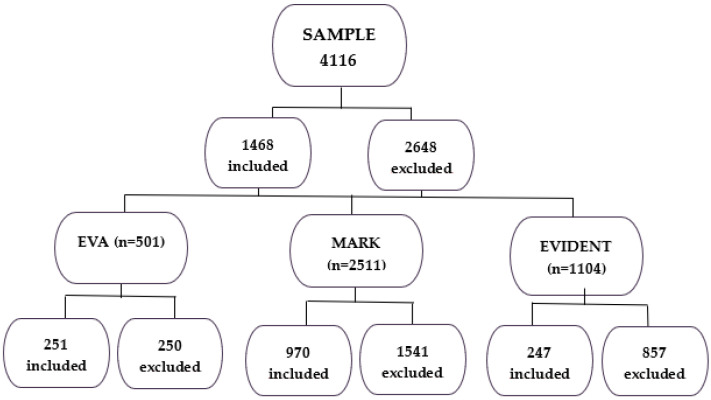
Distribution of subjects included according to study of origin.

**Figure 2 nutrients-17-02963-f002:**
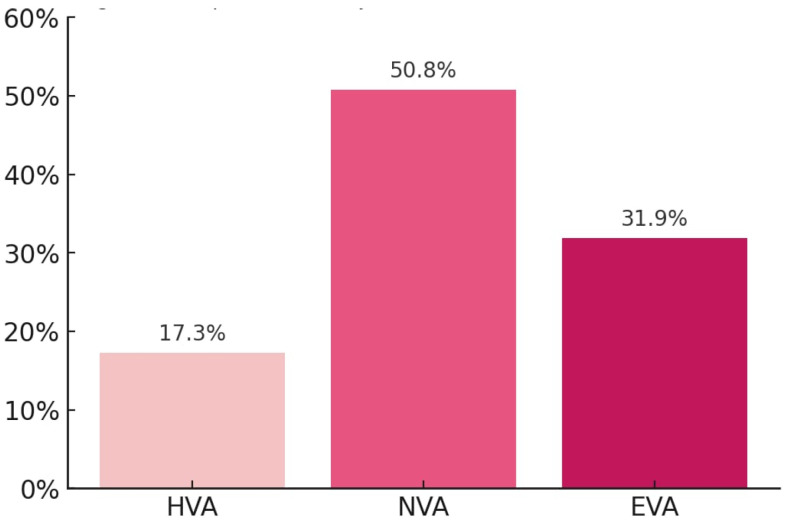
Proportion of subjects with HVA, NVA, and EVA overall. HVA: healthy vascular aging, NVA: normal vascular aging, EVA: early vascular aging.

**Figure 3 nutrients-17-02963-f003:**
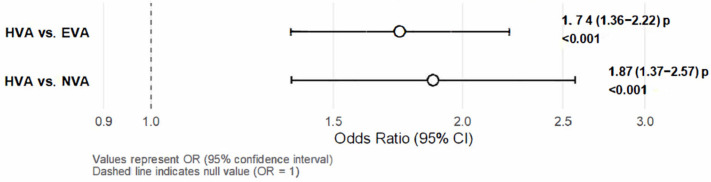
Association of CAVI and MD. Multinomial logistic regression. HVA: healthy vascular aging; NVA: normal vascular aging; EVA: early vascular aging; CAVI: Cardio-Ankle Vascular Index; MD: Mediterranean Diet.

**Table 1 nutrients-17-02963-t001:** General characteristics of the subjects included.

	Total (n = 1468)	30–40 (n = 69)	40–50 (n = 143)	50–60 (n = 434)	60–70 (n = 583)	70–80 (n = 239)	*p* Value
Mediterranean Diet							
MD (total score)	6.03 ± 1.98	6.74 ± 0.24	6.36 ± 0.17	5.89 ± 0.09	5.87 ± 0.08	6.27 ± 0.13	<0.001
Adherence to MD, n (%)	880 (59.9)	31 (44.9)	73 (51.0)	265 (61.1)	377 (64.7)	134 (56.1)	0.001
Conventional risk factors							
Age, (years)	60.09 ± 9.71	35.97 ± 0.30	45.89 ± 0.21	55.51 ± 0.12	64.57 ± 0.10	72.93 ± 0.16	<0.001
SBP, (mmHg)	128.72 ± 20.55	102.73 ± 2.31	117.96 ± 1.60	127.93 ± 0.92	131.07 ± 0.79	135.91 ± 1.24	<0.001
DBP, (mmHg)	79.64 ± 10.84	68.86 ± 1.25	76.23 ± 0.87	82.23 ± 0.50	80.51 ± 0.43	77.98 ± 0.67	<0.001
Hypertension, n (%)	1309 (89.2)	68 (98.6)	134 (93.7)	383 (88.2)	516 (88.5)	208 (87.0)	0.025
Antihypertensive drugs, n (%)	654 (44.6)	1 (1.4)	22 (15.4)	175 (40.3)	291 (49.9)	165 (69.0)	<0.001
Total cholesterol, (mg/dL)	220.54 ± 42.47	182.13 ± 4.94	203.10 ± 3.43	226.35 ± 1.97	226.02 ± 1.70	218.16 ± 2.66	<0.001
LDL cholesterol, (mg/dL)	132.53 ± 35.94	102.74 ± 4.28	122.60 ± 2.94	138.77 ± 1.70	135.61 ± 1.45	128.14 ± 2.27	<0.001
HDL cholesterol, (mg/dL)	57.27 ± 15.77	61.98 ± 1.91	57.16 ± 1.32	56.87 ± 0.76	56.65 ± 0.65	58.20 ± 1.01	0.09
Triglycerides, (mg/dL)	122.63 ± 68.62	76.61 ± 8.18	101.79 ± 5.65	132.05 ± 3.24	125.33 ± 2.80	124.55 ± 4.36	<0.001
Dyslipidemia, n (%)	1076 (73.3)	19 (17.5)	71 (49.7)	317 (73.0)	477 (81.8)	192 (80.3)	<0.001
Lipid–lowering drugs. n (%)	429 (29.2)	1 (1.4)	13 (9.1)	98 (22.6)	207 (35.5)	110 (46.9)	<0.001
FPG, (mg/dL)	100.24 ± 32.30	83.80 ± 3.82	85.64 ± 2.65	102.15 ± 1.53	103.40 ± 1.32	102.58 ± 2.05	<0.001
HbA1c	5.93 ± 1.07	5.26 ± 0.13	5.39 ± 0.09	5.98 ± 0.05	6.07 ± 0.04	6.01 ± 0.07	<0.001
Diabetes mellitus, n (%)	306 (20.8)	2 (2.9)	5 (3.5)	92 (21.2)	147 (25.2)	60 (25.1)	<0.001
Hypoglycemic drugs, n (%)	234 (15.9)	2 (2.9)	4 (2.8)	66 (15.2)	107 (18.4)	55 (23.0)	<0.001
Weight, kg	70.20 ± 13.31	63.91 ± 1.59	70.46 ± 1.10	71.31 ± 0.63	71.03 ± 0.55	67.81 ± 0.85	<0.001
Height, cm	157.11 ± 6.67	162.10 ± 0.76	161.25 ± 0.53	158.24 ± 0.30	155.82 ± 0.26	154.29 ± 0.41	<0.001
BMI, (kg/m^2^)	28.47 ± 5.30	24.35 ± 0.63	27.11 ± 0.43	28.49 ± 0.25	29.27 ± 0.21	28.48 ± 0.34	<0.001
WC, cm	94.33 ± 12.90	82.84 ± 1.51	90.61 ± 1.06	93.61 ± 0.60	96.48 ± 0.52	95.91 ± 0.81	<0.001
Obesity, n (%)	495 (33.7)	9 (13.0)	38 (26.6)	152 (35.0)	213 (36.5)	83 (34.7)	0.001
Arterial stiffness							
CAVI, m/second	8.37 ± 1.24	6.58 ± 0.12	7.18 ± 0.09	8.10 ± 0.05	8.70 ± 0.04	9.29 ± 0.07	<0.001

Values are means ± standard deviations for continuous data and number and proportions for categorical data. MD: Mediterranean Diet; SBP: systolic blood pressure; DBP: diastolic blood pressure; LDL: low–density lipoprotein; HDL: high–density lipoprotein; FPG: fasting plasma glucose; HbA1c: glycosylated hemoglobin; BMI: body mass index; WC: Waist circumference. *p* value: differences between men and women.

**Table 2 nutrients-17-02963-t002:** (**a**) Characteristics of subjects included with and without healthy vascular aging; (**b**) Pairwise comparisons of metabolic and lipid parameters between vascular aging groups (HVA, NVA, EVA).

(**a**)				
	**HVA (n = 254)**	**NVA (n = 746)**	**EVA (n = 468)**	***p* Value**
Mediterranean Diet				
MD (total score)	6.34 ± 0.12	6.12 ± 0.07	5.72 ± 0.09	<0.001
Adherence to MD, n (%)	138 (34.3)	419 (56.2)	323 (69.0)	<0.001
Conventional risk factors				
Age, (years)	58.67 ± 0.61	60.39 ± 0.36	60.38 ± 0.45	0.038
SBP, (mmHg)	124.81 ± 1.27	127.02 ± 0.74	133.54 ± 0.94	<0.001
DBP, (mmHg)	78.43 ± 0.67	78.67 ± 0.39	81.84 ± 0.50	<0.001
Hypertension, n (%)	236 (92.9)	672 (90.1)	401 (85.7)	<0.006
Antihypertensive drugs, n (%)	112 (44.1)	325 (43.6)	217 (46.4)	0.625
Total cholesterol, (mg/dL)	217.47 ± 2.66	218.39 ± 1.55	225.65 ± 1.96	0.007
LDL cholesterol, (mg/dL)	131.33 ± 2.27	131.50 ± 1.32	134.84 ± 1.67	0.246
HDL cholesterol, (mg/dL)	57.85 ± 0.99	57.43 ± 0.58	56.70 ± 0.73	0.598
Triglycerides, (mg/dL)	116.89 ± 4.30	120.62 ± 2.51	128.96 ± 3.17	<0.041
Dyslipidemia, n (%)	165 (65.0)	536 (71.8)	375 (80.1)	<0.001
Lipid–lowering drugs. n (%)	61 (24.0)	220 (29.5)	148 (31.6)	0.097
FPG, (mg/dL)	88.54 ± 2.00	100.10 ± 1.16	106.79 ± 1.47	<0.001
HbA1c	5.47 ± 0.07	5.94 ± 0.04	6.16 ± 0.05	<0.001
Diabetes mellitus, n (%)	0 (0.0)	172 (23.1)	134 (28.6)	<0.001
Hypoglycemic drugs, n (%)	0 (0.0)	133 (17.8)	101 (21.6)	<0.001
Weight, kg	73.85 ± 0.83	70.05 ± 0.48	68.45 ± 0.61	<0.001
Height, cm	156.68 ± 0.42	154.86 ± 0.24	157.75 ± 0.31	<0.004
BMI, (kg/m^2^)	30.08 ± 0.33	28.53 ± 0.19	27.52 ± 0.24	<0.001
WC, cm	95.49 ± 0.81	94.22 ± 0.47	93.87 ± 0.60	0.262
Obesity, n (%)	11 (43.7)	259 (34.7)	125 (26.7)	<0.001
Arterial stiffness				
CAVI, m/second	6.95 ± 0.06	8.19 ± 0.03	9.44 ± 0.04	<0.001
(**b**)				
	**HVA vs. NVA**	**HVA vs. EVA**	**NVA vs. EVA**	
Mediterranean Diet				
Total cholesterol, (mg/dL)	0.767	0.013	0.004	
Triglycerides, (mg/dL)	0.454	0.024	0.039	
FPG, (mg/dL)	<0.001	<0.001	<0.001	
HbA1c	<0.001	<0.001	<0.001	

Values are means ± standard deviations for continuous data and number and proportions for categorical data. MD: Mediterranean Diet; SBP: systolic blood pressure; DBP: diastolic blood pressure; LDL: low–density lipoprotein; HDL: high–density lipoprotein; FPG: fasting plasma glucose; HbA1c: glycosylated hemoglobin; BMI: body mass index; WC: Waist circumference; HVA: healthy vascular aging; NVA: normal vascular aging; EVA: early vascular aging; *p* value: differences between groups.

## Data Availability

The data supporting the findings of this study are available on ZENODO under the DOI: https://doi.org/10.5281/zenodo.12166167.

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
