# Peer review of "Influence of Adherence to the Mediterranean Diet on Vascular Health and Healthy Aging in Women: Assessment Using CAVI"

_nutrients, 2025, doi:10.3390/nu17182963_

Round 1

Reviewer 1 Report

Comments and Suggestions for Authors

This manuscript investigates the association between adherence to the Mediterranean diet and vascular aging in women, assessed by the Cardio-Ankle Vascular Index (CAVI). The study is timely, relevant, and supported by a large sample size. The findings add valuable evidence on the role of dietary patterns in cardiovascular aging, with clear implications for prevention strategies. The paper is generally well written and methodologically sound. However, several issues require clarification and refinement.

  • The rationale for reclassifying women with diabetes from the HVA group to the NVA group should be explained in greater detail. This decision may influence interpretation of vascular aging distribution and should be more clearly justified.
  • The paradoxical finding that a higher proportion of women in the EVA group reported adherence to the MD, despite lower diet scores, needs clearer explanation. The authors should expand on whether this reflects late adoption of dietary changes after diagnosis or other factors.
  • The limitations should be discussed more explicitly. The cross-sectional design precludes causal inference, and reliance on self-reported dietary questionnaires introduces potential recall/reporting bias. These should be highlighted more strongly in the discussion.
  • In section 2.5, "DM adherence" should be corrected to "MD adherence."

  • Table 2 includes the untranslated Spanish phrase “Rigidez arterial.” Please standardize to English.

  • Figure 3 would be improved by including confidence intervals or clearer labeling of effect sizes.

Author Response

Y es

Reviewer 2 Report

Comments and Suggestions for Authors

In this paper, aiming to analyze the association between adherence to the Mediterranean diet and vascular aging estimated with CAVI, a cross-sectional study was conducted with 1468 Spanish women (aged 30–80 years) with no history of previous cardiovascular disease. Cardiovascular status were classified into three categories according to CAVI: healthy vascular aging (HVA), normal (NVA), and early (EVA). Adherence to the Mediterranean diet was assessed using the MEDAS questionnaire validated in Spain. Statistical analysis performed and authors conclude that greater adherence to the MD is associated with a healthier vascular aging profile.

Here are some concerns:

  • Recently there is a review paper on the effect of a Mediterranean Diet on arterial stiffness, 128 publications were evaluated, 16 observational and randomized controlled trials that aligned with the research question were reviewed. The evidence showed an inverse relationship between adherence to the Mediterranean diet and arterial stiffness, with a focus on pulse wave velocity (PWV) and the Augmentation Index (AIx) as outcome measures. Lower but consistent and statistically significant evidence was also found in the cross-tabulation of adherence to the Mediterranean diet and the cardiovascular ankle index (CAVI), a proxy of the overall stiffness of the artery from the origin of the aorta to the ankle. The authors should take into consideration of this review in their discussion.

(The Effect of a Mediterranean Diet on Arterial Stiffness: A Systematic Review. Nutrients 2025, 17, 1192).

  • There were studies investigating the increase of CAVI with aging trying to figure out a pattern. The authors should discuss these works.
  • Namekata T., Suzuki K., Ishizuka N., Shirai K. Establishing Baseline Criteria of Cardio-Ankle Vascular Index as a New Indicator of Arteriosclerosis: A Cross-Sectional Study. BMC Cardiovasc. Disord. 2011;11:51. doi: 10.1186/1471-2261-11-51. [DOI] [PMC free article] [PubMed] [Google Scholar]
  • Shirai K. Analysis of Vascular Function Using the Cardio–Ankle Vascular Index (CAVI) Hypertens. Res. 2011;34:684–685. doi: 10.1038/hr.2011.40. [DOI] [PubMed] [Google Scholar]

  • The paper needs to be carefully reviewed about its English, there are places Spanish was used in the text and section title.

  • Specific concerns:

Line 87, “ notoriety” by definition, it means “the state of being famous or well known for some bad quality or deed.”

Is this what the authors mean?  Or the authors mean “reputation”

Line 197: “DM adherence”, please define DM, what does this stand for?

Line 228, please use English in the writing.

Characteristics of the subjects included with and without healthy vascular aging.

Line 229-235: “Women were classified into healthy vascular aging (HVA, n=254), normal aging  (NVA, n=746), and early vascular aging (EVA, n=468). We found a better score for MD  adherence in the HVA group (6.34 ± 0.12) compared to the EVA group (5.72 ± 0.09) (p <  0.001). On the other hand, the proportion of subjects with high adherence to the MD was  higher in the EVA group (69.0%) than in the HVA group (34.3%), which could reflect greater implementation of dietary changes in subjects with higher cardiovascular risk (p < 0.001)”

I don’t understand why the difference can reflect the greater implementation of dietary changes in subjects with higher cardiovascular risk. Can you present data for the MD score for the patient at the beginning and the end of the study? This might be a better parameter to reflect the dietary changes for the different groups.

Table 1 and Table 2:”CAVI m/Segundo”,  please make sure that English is used in the whole paper.

Table 1, There were studies investigating the increase of CAVI with aging trying to figure out a pattern. The authors should discuss these works.

  • Namekata T., Suzuki K., Ishizuka N., Shirai K. Establishing Baseline Criteria of Cardio-Ankle Vascular Index as a New Indicator of Arteriosclerosis: A Cross-Sectional Study. BMC Cardiovasc. Disord. 2011;11:51. doi: 10.1186/1471-2261-11-51. [DOI] [PMC free article] [PubMed] [Google Scholar]
  • Shirai K. Analysis of Vascular Function Using the Cardio–Ankle Vascular Index (CAVI) Hypertens. Res. 2011;34:684–685. doi: 10.1038/hr.2011.40. [DOI] [PubMed] [Google Scholar]

For Table 2, lines236-241. It would be better for the authors to also point out the differences seen in the total cholesterol, triglycerides, FPG, HbA1c for the three groups.

Comments on the Quality of English Language

The paper needs to be carefully reviewed about its English, there are places Spanish was used in the text and section title.

Author Response

Y Es

Round 2

Reviewer 2 Report

Comments and Suggestions for Authors

The authors have done a good job in the revision.  There are just some minor problems need s attention:

Line 90:  "Reputation" is a lot better, another word that can be used here is "recognition".  Really depend on what message the authors want to convey.

Line 382:  Association between CAVI y MD overall and by sex

Association between CAVI and MD overall and by sex

Comments on the Quality of English Language

good now

Author Response

Dear Reviewer 2.

Thank you for your comments and suggestions. We are very grateful for the time you have spent reviewing our manuscript and, above all, for the quality it has acquired.

Line 90:  "Reputation" is a lot better, another word that can be used here is "recognition".  Really depend on what message the authors want to convey.

We have changed the word.

Line 382:  Association between CAVI y MD overall and by sex

Association between CAVI and MD overall and by sex

We apologize for the mistake, which has now been corrected and translated correctly.
